# Origin of two-band chorus in the radiation belt of Earth

Jinxing Li [1]*, Jacob Bortnik[1]*, Xin An [1], Wen Li[2], Vassilis Angelopoulos [3], Richard M. Thorne[1], Christopher T. Russell[3], Binbin Ni[4,5], Xiaochen Shen[2], William S. Kurth [6], George B. Hospodarsky[6], David P. Hartley [6], Herbert O. Funsten [7], Harlan E. Spence[8] & Daniel N. Baker[9]

Naturally occurring chorus emissions are a class of electromagnetic waves found in the space environments of the Earth and other magnetized planets. They play an essential role in accelerating high-energy electrons forming the hazardous radiation belt environment. Chorus typically occurs in two distinct frequency bands separated by a gap. The origin of this two-band structure remains a 50-year old question. Here we report, using NASA's Van Allen Probe measurements, that banded chorus waves are commonly accompanied by two separate anisotropic electron components. Using numerical simulations, we show that the initially excited single-band chorus waves alter the electron distribution immediately via Landau resonance, and suppress the electron anisotropy at medium energies. This naturally divides the electron anisotropy into a low and a high energy components which excite the upper-band and lower-band chorus waves, respectively. This mechanism may also apply to the generation of chorus waves in other magnetized planetary magnetospheres.

[1] Department of Atmospheric and Oceanic Sciences, University of California, Los Angeles, CA 90095, USA. [2] Center for Space Physics, Boston University, Boston, MA, USA. [3] Department of Earth, Space and Planetary Sciences, University of California, Los Angeles, CA 90095, USA. [4] Department of Space Physics, School of Electronic Information, Wuhan University, Wuhan, Hubei 430072, China. [5] CAS Center for Excellence in Comparative Planetology, Heifei, Anhui 230026, China. [6] Department of Physics and Astronomy, University of Iowa, Iowa City, IA 52242-1479, USA. [7] Los Alamos National Laboratory, MS-D466, PO Box 1663, Los Alamos, NM 87545, USA. [8] Institute for the Study of Earth, Oceans, and Space, University of New Hampshire, Durham, NH 03824-3525, USA. [9] Laboratory for Atmospheric and Space Physics, University of Colorado, Boulder, CO, USA. *email: jinxing.li.87@gmail.com; jbortnik@gmail.com

Whistler mode chorus waves are a class of naturally occurring coherent electromagnetic emissions found commonly in the Earth's near-space magnetic environment known as the magnetosphere, as well as other planetary magnetospheres[1–3]. They are named chorus waves because they sound like the dawn chorus of birds when converted to audio[4,5]. They play a key role in forming a zone consisting of relativistic electrons, known as Earth's outer radiation belt[6,7] (Fig. 1a), and in precipitating energetic (0.1–30 keV) electrons to the upper atmosphere to produce diffuse aurora[8–10] and pulsating aurora[11,12]. They are also an important source of hiss waves which can deplete the electrons in the slot region between the inner and outer radiation belts[13–16]. Chorus waves in the Earth's magnetosphere typically occur in two distinct frequency bands: a lower-band (0.1–0.5 $f_{ce}$) and an upper-band (0.5–0.8 $f_{ce}$), where $f_{ce}$ represents the equatorial electron gyro-frequency[1,17–19]. This two-band structure occurs both at the magnetic equator[20,21] and

in off-equatorial regions. It also occurs in the magnetospheres of Jupiter[22–24], Saturn[25,26], and Mars[27].

Numerous mechanisms have been suggested to explain the banded nature of chorus waves over the past few decades, with limited success. For example, Landau damping at ~0.5 $f_{ce}$ at high latitudes[18,28], and nonlinear longitudinal damping due to the inhomogeneity of the static magnetic field[29,30], have been invoked but neither can explain the banded chorus waves observed near the source region, the magnetic equator. It has been proposed that the lower-band can be triggered by the upper-band[31] (or vice versa[32]) via a nonlinear wave–wave coupling, but this theory cannot explain general cases in which the two bands have no correlation. It has been suggested that the upper- and lower-band chorus waves originate from two separate anisotropic electron populations, while a close-to-isotropic component at medium energies suppresses the chorus wave growth at 0.5 $f_{ce}$[33,34]. However, it is unknown whether this causality is general or just a

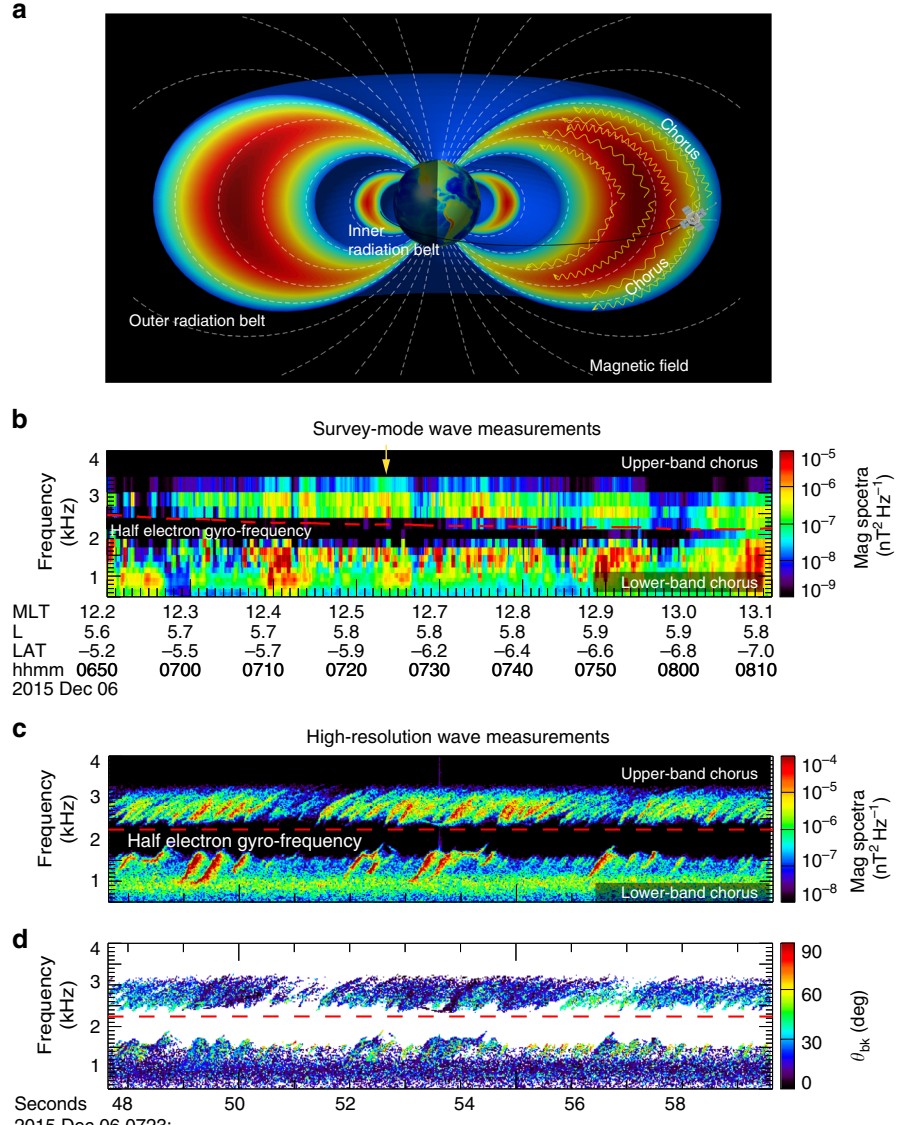

**Fig. 1** Van Allen Probe-A orbit and observations of two-band chorus waves. **a** The configuration of outer and inner radiation belts that are confined by Earth's magnetic field (the colors are not from any specific data), as well as the Van Allen Probe orbit and its position at 07:23 UT on 6 December 2015. **b** 80-min overview of survey-mode magnetic wave spectral intensity, showing two-band chorus waves separated by a gap around half electron gyro-frequency, which is represented by a dashed red line. **c** 12-second high-resolution measurements of magnetic wave spectral intensity during the time indicated by a yellow arrow in Fig. 1b, showing discrete rising-tone elements in each band. **d** The wave normal angles calculated from the waveform data using the Means method[41], showing that the chorus waves propagate mostly at wave normal angles of 0º –30º with respect to the magnetic field

coincidence in one specific case, and how the subtle multi-component electron distribution is developed. A recent particle-in-cell simulation[35] demonstrated growth of two-band chorus waves resulting from two anisotropic electron populations, but did not address how the electron population developed into two anisotropic components. Other proposed explanations, including different source regions[36], ducting by plasma density structures[37], have had limited observational support. Thus, there is currently no consensus for the origin of banded chorus.

Understanding the structure and formation of the chorus spectrum is of great importance for understanding and predicting the physical processes responsible for electron acceleration and loss in the radiation belts. Here we show, by presenting Van Allen Probe[38] observations, that two-band chorus waves are usually accompanied by electron parallel acceleration and two separate anisotropic electron components. We show a particle-in-cell simulation to demonstrate the generation mechanism of two-band chorus waves and the evolution of electron distributions in the process.

## Results

**Chorus wave measurements**. During a moderate geomagnetic storm on 6 December 2015, Van Allen Probe-A captured dayside chorus waves in the outer radiation belt, at an altitude of ~30,000 km (L~6, where L is the maximum radial extent of the dipole magnetic field in units of Earth Radii) from the Earth and a magnetic latitude (LAT) of about −6.0° (Fig. 1a, b). The 80-minute survey-mode magnetic wave dynamic spectrum measured by the waveform receiver instrument[39] shows two-band chorus waves separated in frequency by a gap at 0.5 $f_{ce}$. Here $f_{ce}$ is determined by the magnetic field strength $B$ as $f_{ce} = eB/2\pi m$, where $e$ is the elementary charge, $m$ is the electron mass. The high-cadence wave dynamic spectrum, shown in Fig. 1c, demonstrates that both bands consist of discrete rising-tone elements[1,40], which grow separately in each band. The banded chorus can be heard by converting the waveform to audio (Supplementary Video 1). The wave normal angles calculated from the high-cadence data using the Means method[41], shown in Fig. 1d, indicate that both bands propagate at angles of 0°–30° with respect to the geomagnetic field, which can be considered roughly field-aligned.

**Electron measurements and linear growth analysis**. Electrons and chorus waves exchange energy primarily via two types of resonant interactions: cyclotron resonance, which occurs when the Doppler-shifted wave frequency in the particle's moving frame matches its gyro-frequency, $\omega - k_{\parallel} v_c = \omega_{ce}$, and Landau resonance, which occurs when the wave is stationary in the particle's frame, $\omega - k_{\parallel} v_L = 0$. Here $\omega = 2\pi f$ is the angular wave frequency, $k_{\parallel}$ the parallel wave number, $v_c$ the cyclotron resonant parallel velocity and $v_L$ the Landau resonant parallel velocity (where parallel refers to the direction of the background magnetic field). At the frequency of 0.5 $f_{ce}$, the magnitudes of the two velocities are equal, $|v_c| = |v_L|$.

It has been well known that chorus waves are generally excited by injected electrons whose perpendicular temperature (with respect to the background magnetic field) exceeds the parallel temperature, and are excited via cyclotron resonant interactions[42,43]. Figure 2a shows 15-min averaged electron phase space density (PSD) profiles at different local pitch angles measured by the Helium, Oxygen, Proton and Electron (HOPE) Spectrometer[44,45]. The black line represents the PSD of ~18° local pitch angles and the red line represents the PSD measured at ~90°

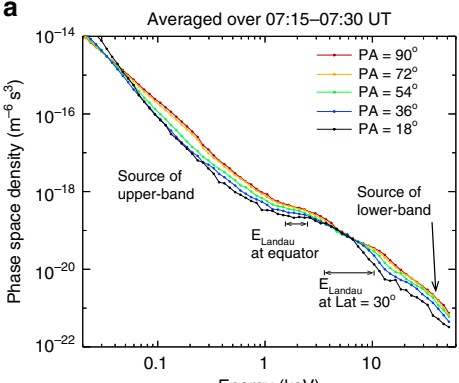

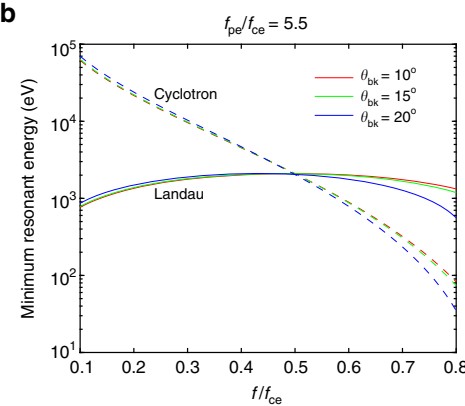

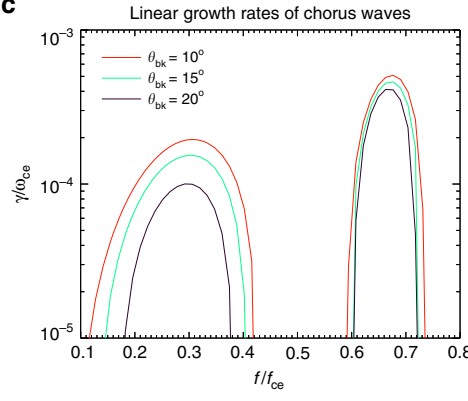

**Fig. 2** Electron PSD measurements, resonant energy analysis and linear growth calculations. **a** 15-minute-averaged electron PSD versus energy measured by the HOPE spectrometer onboard the Van Allen Probe. Electrons exhibit significant anisotropies over energies of 0.05–2 keV and again at >10 keV, providing energy sources to the upper-band and lower-band chorus waves, respectively. A parallel PSD plateau is seen at ~2 keV, around the equatorial Landau resonant energies. The PSD of 4–6 keV at 18º exceeds the PSDs at 36º and 54º pitch angles, suggesting an acceleration in the parallel direction, possibly caused by Landau resonant accelerations at higher latitudes. **b** Minimum resonant energies for cyclotron and Landau resonances as a function of wave frequency for various wave normal angles under the plasma condition of $f_{pe}/f_{ce} = 5.5$. The cyclotron resonance occurs over a broad energy range from 0.22 to 24 keV, while the Landau resonant energy is within a small energy range from 1.3 to 2.1 keV. The equatorial Landau resonant energies match the observed PSD plateau. **c** Chorus wave growth rate ($\gamma/\omega_{ce}$) as a function of frequency for three selected wave normal angles, indicating that chorus waves can grow at the lower and upper bands

**Table 1 The fitting parameters for the electron distribution shown in Fig. 2a**

| Type | $Ne$ (cm$^{-3}$) | $v_{\perp}$ (km/s) | $v_{\parallel}$ (km/s) | $v_{beam}$ (km/s) |
|---|---|---|---|---|
| Kappa, $\kappa = 4$ | 7.3 | 1000 | 1200 | 0 |
| Kappa, $\kappa = 2$ | 0.6 | 3000 | 2000 | 0 |
| Gaussian | 0.03 | 30,000 | 8000 | 0 |
| Gaussian | 0.0024 | 20,000 | 10,000 | 32,000 |
| Kappa, $\kappa = 2$ | 0.064 | 40,000 | 30,000 | 0 |

The κ index determines the slop of the Kappa distribution at the tail

local pitch angle (77° equatorial pitch angle). Two distinct anisotropic electron populations can be seen, with perpendicular (90° pitch angle) PSD larger than parallel (18° pitch angle) PSD, at energies of 0.05–2 keV and above 10 keV, respectively. At energies between these two anisotropic populations, the electrons are more isotropic. The minimum resonant electron energies (velocities) versus chorus frequency calculated using cold plasma theory[46], shown in Fig. 2b, indicate that the two anisotropic components are in the cyclotron resonant energy ranges of the upper-band and lower-band chorus waves, respectively.

To numerically evaluate the wave growth from the anisotropy instability, we perform a linear growth calculation[42] which describes the initial phase of wave excitation. Following a similar procedure described in previous studies[47–49], we first fit the measured electron PSD distributions as a summation of bi-Maxwellian distribution and Kappa (power law) distribution[50] (See "Methods" and Table 1). With the analytically fitted distribution function, the linear growth rates of whistler mode chorus are calculated as the summation of contributions caused by the cyclotron resonance and Landau resonance. The overall linear growth rates, shown in Fig. 2c, demonstrate that the chorus waves can grow separately at both the upper and lower bands, while the chorus waves around 0.5 $f_{ce}$ cannot be excited due to insufficient anisotropy at energies of several keV.

**Formation of two separate anisotropic components**. When plasmasheet electrons are injected into the outer radiation belt, their perpendicular energies, which are proportional to the geomagnetic field strength, increase and exceed their parallel energies, forming anisotropic populations typically over 0.1–100 keV. The key question concerning the banded chorus waves is why electrons develop such a distinctive distribution with two anisotropic energy ranges separated by a more isotropic gap that can suppress wave excitation around 0.5 $f_{ce}$.

For chorus waves with frequencies of 0.2–0.7 $f_{ce}$ and wave normal angles of 0°–30°, cyclotron resonance occurs over a broad velocity range, however, Landau resonance occurs over a much narrower velocity range close to 0.5 $v_{Ae}$ (Fig. 2b). Here $v_{Ae}$ is the electron Alfvenic velocity defined as $v_{Ae} = B/\sqrt{\mu_0 m_e n} = c f_{ce}/f_{pe}$, where $n$ is the plasma number density, $\mu_0$ the vacuum magnetic permeability, $c$ is the speed of light, and $f_{pe}$ is the electron plasma frequency. In this case, $f_{ce} = 4.5$ kHz, and $f_{pe} = 25$ kHz, which is determined from the upper hybrid frequency $f_{UH}$ measurements (see Supplementary Fig. 1) via the relation $f_{pe}^2 = f_{UH}^2 - f_{ce}^2$. We note that the parallel electron PSD shows a plateau around 2 keV, roughly at the equatorial Landau resonant energy range of the observed chorus waves. This is similar to PSD plateaus due to Landau resonant electron acceleration by chorus waves, mostly occurring in the lower-band[51]. Moreover, at energies of 4–6 keV, the electron PSD measured at a pitch angle of 18° exceeds the PSD at 36° and 54°, clearly indicating an acceleration process in the direction parallel to the background magnetic field. That acceleration is likely caused by Landau resonance with chorus waves at higher latitudes, because

the Landau resonant velocities, ~0.5 $v_{Ae}$, increase with latitude ($B \propto \sqrt{1 + 3\sin^2\lambda}/\cos^6\lambda$, and $\sqrt{n} \propto 1/\cos^\alpha\lambda$ where $\alpha = 0.8 - 2.1$ according to an empirical model[52], therefore $v_{Ae} \propto B/\sqrt{n}$ increases with latitude roughly as $1/\cos^4\lambda$ or faster). When the chorus waves propagate to a latitude of 30°, they can accelerate electrons up to 10 keV in the parallel direction. The electron parallel acceleration above 10 keV is generally not evident, possibly because the chorus waves have been significantly damped by the time they reach this point and the local normalized frequency, $f/f_{ce}$, becomes very small at high latitudes.

**Statistical survey**. To determine whether Landau resonant acceleration coexists with two-band chorus waves, we analyze Van Allen Probe-A data from January 2013 to October 2014, during which the spacecraft covered all magnetic local times (MLT). We select only those events in which well-defined two-band chorus waves are observed continuously for at least 5 min, and the plasma density can be reliably determined from the upper hybrid frequency. Same-day observations are counted as one event. Using these criteria, 20 events are identified (Supplementary Fig. 2). Figure 3 shows electron distributions for each event. Arrows denote the representative Landau resonant energy for field-aligned electrons, calculated as: $mv_{Ae}^2/8$ (see "Methods"). In all cases the anisotropy is suppressed near this resonant energy, separating the distribution into two anisotropic energy ranges. In 18 out of the 20 cases, parallel electron acceleration, indicated by a PSD plateau or $f(18°) > f(36°)$, is evident. The two exceptions are the 04/19/2014 and 04/21/2014 events, possibly because the Landau resonant acceleration and the formation of two-band chorus waves are still in process.

In many two-band chorus wave cases, the upper hybrid frequency is either below the instrument's effective threshold or confounded by some high-frequency emissions such as electron cyclotron harmonic waves, thus, the plasma density and hence the Landau resonant velocity cannot be reliably determined. Those events, however, often show two anisotropic components as well (see Supplementary Discussion 1 and Supplementary Fig. 3).

**Self-consistent particle-in-cell simulations**. To numerically simulate the evolution of the electron distribution and the generation of two-band chorus waves, we perform a self-consistent particle-in-cell simulation. We use the Darwin code[53], which neglects the transverse displacement current but leaves the physics of whistler mode waves unaffected. It removes the time-step constraint of the full electromagnetic code, and the computational efficiency can be greatly improved by using a large time-step. The Darwin code had been successful in explaining the whistler mode wave excitation and the resultant electron distributions both in space and laboratory plasma environments[54,55]. The one-dimensional Darwin model has one spatial dimension in the longitudinal direction $x$ (the particles and fields have no variations along the transverse directions) and three dimensions in particle velocity. This geometry allows waves to propagate only in the specified longitudinal direction. In our simulation, the background magnetic field is uniform, and the wave normal angle is set at $\theta_{bk} = 15°$. The $f_{pe}/f_{ce}$ ratio is 5 consistent with regions where chorus is generated in the magnetosphere. The measured electron PSD in the present case has a long tail at high energies and is better approximated by a Kappa distribution than a Maxwellian. We thus model the initial electron distribution as a combination of isotropic cold and anisotropic warm components, both of which are described by the Kappa distributions. The parameters of each population are shown in Table 2. We set the electron anisotropy in the simulation to be larger than the one measured in order to speed up the wave growth process (a common treatment in such numerical simulations). A total of ~8 million

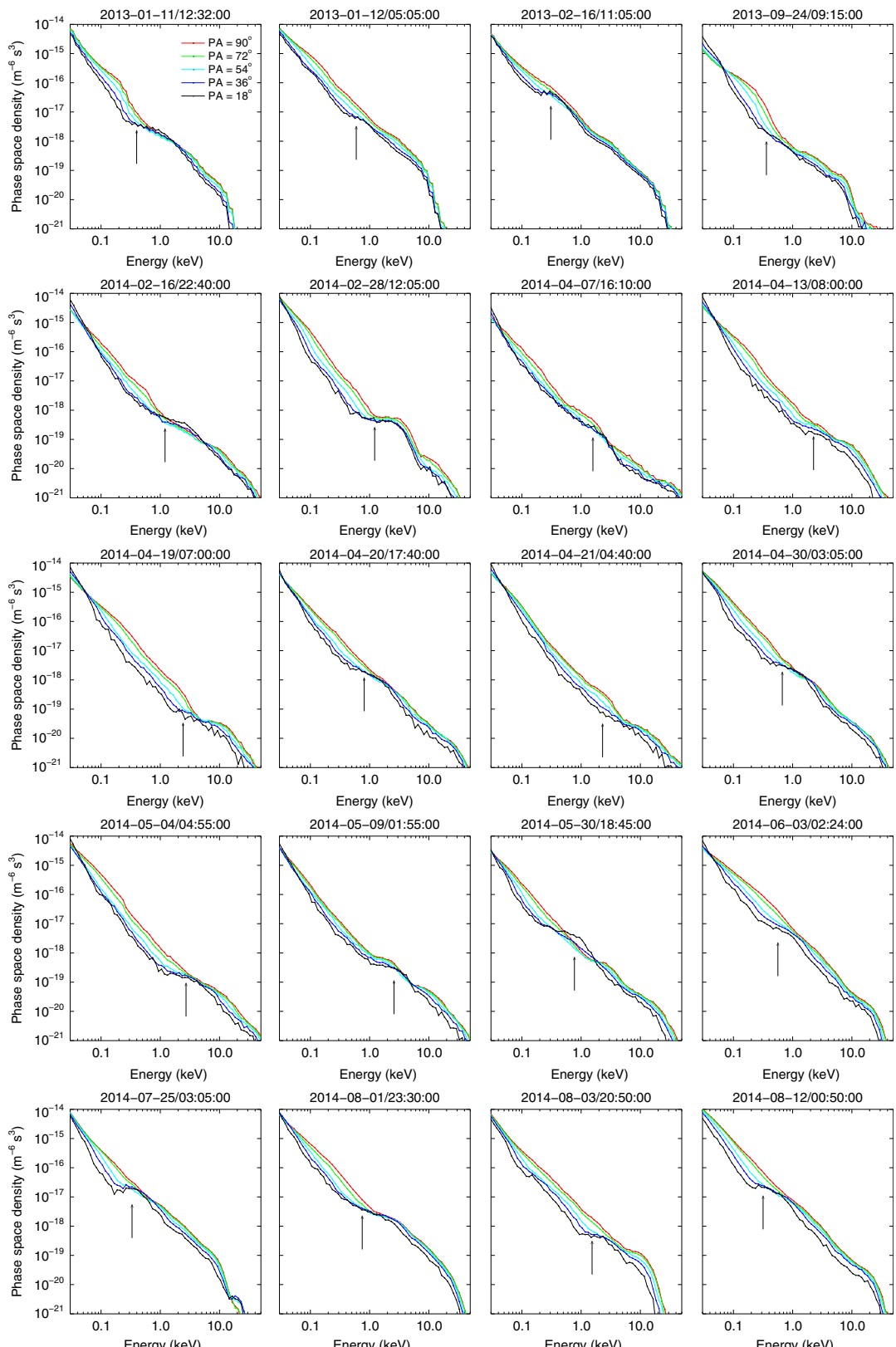

**Fig. 3** Electron phase space densities during the 20 two-band chorus wave events. The arrows denote the representative equatorial Landau resonant energy. The color legends in Figure a are valid for the entire figure. Eighteen events show clear evidence of parallel acceleration at Landau resonant energies, identified by either electron parallel PSD plateau or that the PSD at 18° exceeds the PSD at 36° and even 54° pitch angles. The two exceptions are the 2014-04-19 and 2014-04-21 events where the parallel accelerations are not evident. However, all those 20 events show two anisotropic components separated by the Landau resonant energies

electrons are spatially distributed into the 1024 longitudinal cells, and the initial PSD distribution is illustrated in Fig. 4a. We use a periodic boundary condition and neglect the loss cone effect in the simulation. The time-step is $0.032 f_{pe}^{-1}$.

At the beginning of the simulation, the chorus waves grow through cyclotron resonant interactions with the well-defined anisotropic electrons, and are excited in one continuous band over the frequency range of 0.2–0.8 $f_{ce}$ with the growth at the upper-band proceeding faster as seen in Fig. 4b. Note that the wave spectra are shown as a function of frequency ($\omega$) and $k$ mode number (the number of wavelengths in the simulation space), and do not represent the rising-tone structures commonly observed in frequency-time spectrograms. The chorus waves propagate along the +x and −x directions evenly due to the symmetric electron velocity distribution, which is similar to the scenario at the geomagnetic equator. As the one-band chorus waves intensify, they

interact with electrons via Landau resonance and accelerate electrons in the parallel direction[48,51], thus reducing the electron anisotropy at the Landau resonant energies ~ 2–3 keV (Fig. 4c). This energy range corresponds to the minimum cyclotron resonance energy of chorus waves around 0.5 $f_{ce}$ (Fig. 2c), therefore, the chorus waves at 0.5 $f_{ce}$ become notably weaker and an upper- and lower-band structure starts to develop (Fig. 4d). As the chorus waves continue to grow and accelerate electrons via Landau resonance, a pronounced electron PSD plateau is formed around the Landau resonant energy range, which separates the anisotropic electrons into two parts (Fig. 4e). As a result, as illustrated in Fig. 4f, the chorus wave growth is significantly suppressed at ~0.5 $f_{ce}$ but remains active at the lower and upper bands, thereby producing the two-band structure of chorus waves.

## Discussion

In this 1-D simulation, we set the wave propagation angle to be 15° to represent the measured range of 0°-30°. In the magnetosphere, the quasi-parallel propagating chorus waves statistically have a normal angle distribution centered around ~10°–15°[20]. It is important to note that the wave normal angles typically change by tens of degrees even within a single chorus subpacket[56]. Banded chorus waves with a small central-wave normal angles (e.g., 8°) are also observed. Those waves too exhibit oblique normal angles over a large portion of their wavepacket structure[21]. More importantly, due to the geomagnetic dipole configuration, chorus waves generated at equator, even with very small initial central-wave normal

**Table 2 The kappa distribution parameters in the model**

|  | Distribution type | Percentage | $v_{T\parallel}$ | $v_{T\perp}$ |
|---|---|---|---|---|
| Cold | $\kappa = 4$ | 80% | 0.01c | 0.01c |
| Warm | $\kappa = 1.5$ | 20% | 0.02c | 0.04c |

The electron initial distribution is a combination of a cold isotropic and a warm anisotropic components

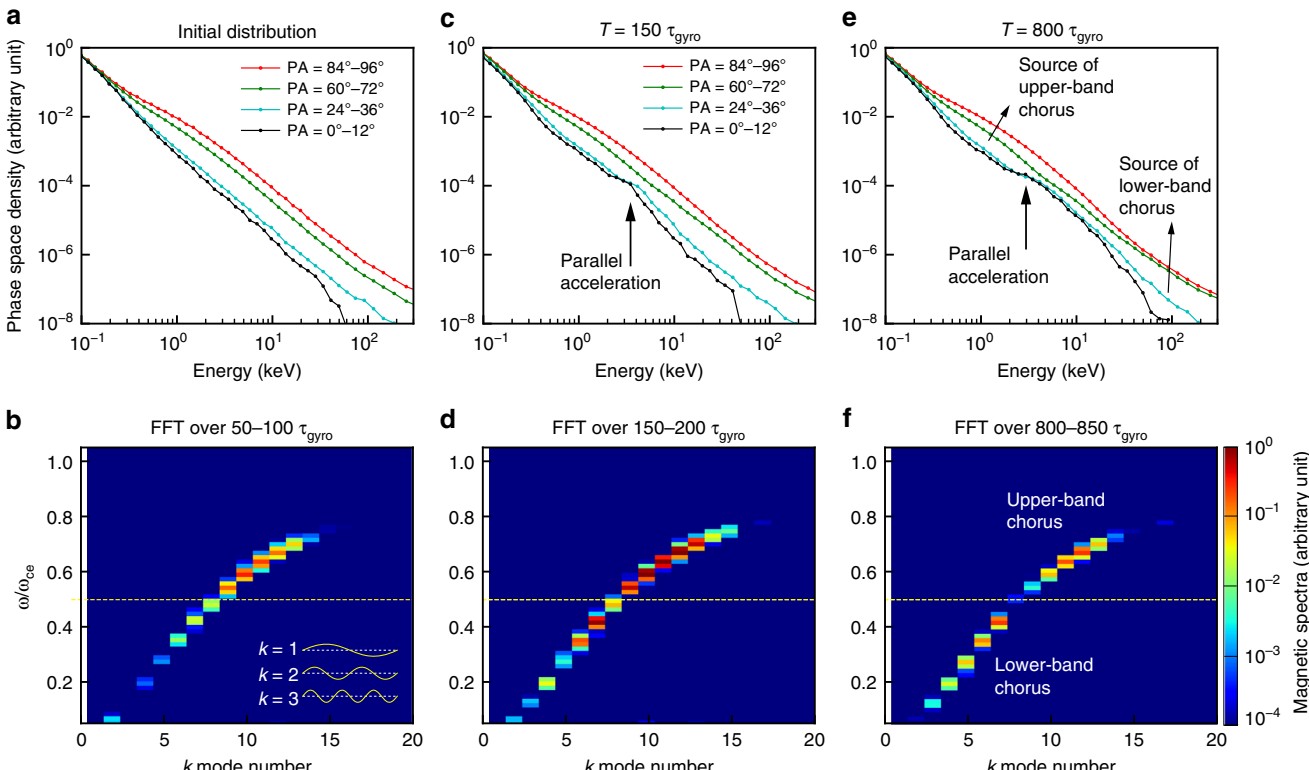

**Fig. 4** Simulation of chorus wave excitation and the concurrent electron PSD variation. **a** The initial electron PSD versus energy at selected pitch angles, showing a cold isotropic population extending to ~100 eV and a warm anisotropic population above ~ 300 eV. **b** The wave magnetic spectral intensity for each ($\omega$, $k$) mode obtained by Fourier transformation of the magnetic field over $T = 50$–$100\ \tau_{gyro}$ (electron gyro-period), showing a continuous one-band spectral structure, while the excitation of upper-band chorus is faster. **c** The electron PSD at $T = 150\ \tau_{gyro}$, showing a notable parallel acceleration at ~2–3 keV, which is around the Landau resonant energies of the chorus waves. **d** As the chorus waves at both bands continues to grow, the wave intensity at 0.5 $f_{ce}$ is notably weaker than the two bands, due to the reduce anisotropy at ~2–3 keV. **e** At $T = 800\ \tau_{gyro}$, the electron PSD exhibits two distinct anisotropic components, separated by the electrons that have undergone parallel acceleration by chorus waves. **f** The resultant two-band chorus waves with a gap at 0.5 $f_{ce}$. The upper-band chorus grows from the ~1 keV anisotropic component, and the lower-band chorus grows from tens of keV anisotropic component

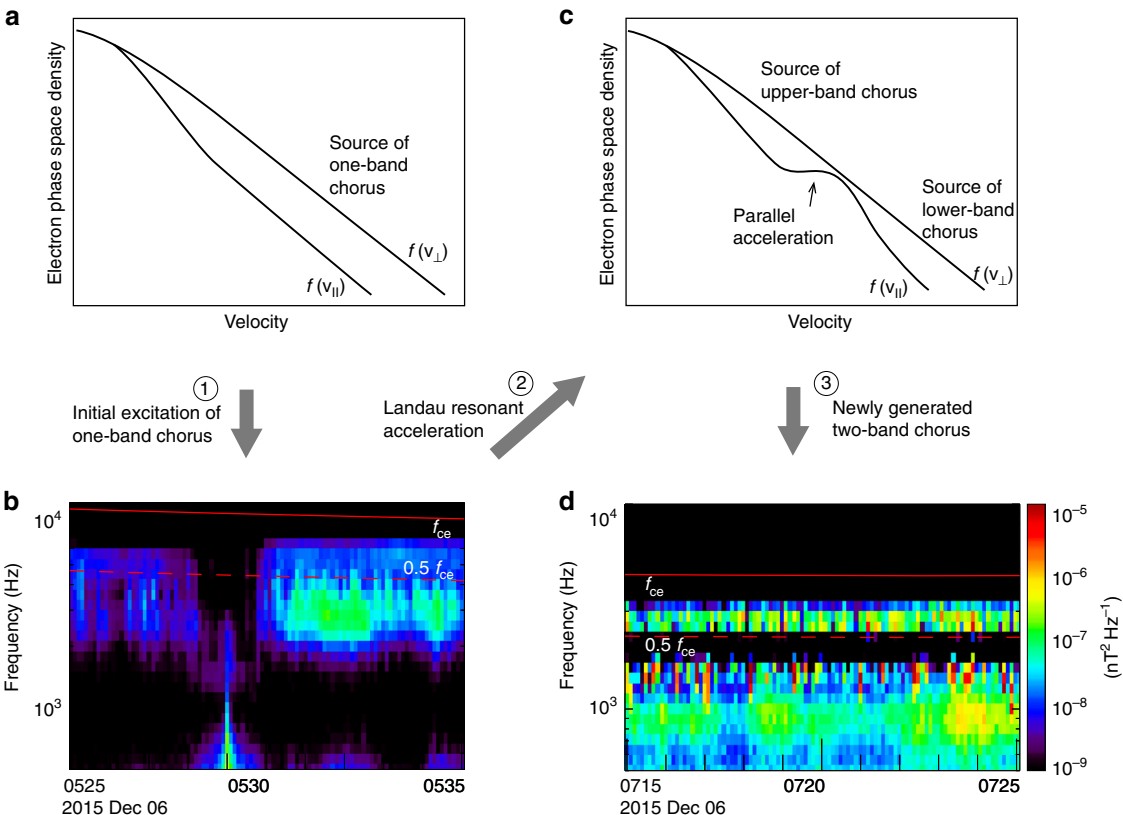

**Fig. 5** Schematic illustration of the generation mechanism of two-band chorus waves. **a** The anisotropic velocity distribution of freshly injected electrons and **b** the generation of one-band chorus emissions. **c** The electron distribution after Landau resonant acceleration taking place, resulting in two separate anisotropic parts, **d** The excitation of the upper and the lower band chorus emissions by the low-energy and high-energy anisotropic electrons, respectively. The red solid and dashed lines represent $f_{ce}$ and 0.5 $f_{ce}$, respectively. Since the Landau resonant acceleration and the formation of two anisotropic components occur very rapidly, the two-band chorus waves are more commonly observed than the one-band chorus waves. The measured electron distribution associated with the one-band and two-band chorus waves are shown in Supplementary Fig. 6

angles, can become oblique when they propagate to higher latitudes[14] and cause pronounced Landau acceleration there and the resultant two anisotropic components over the wave-particle interaction volume. While most chorus waves have small wave normal angles[57], banded chorus waves with a quasi-electrostatic, very oblique lower band is also observed[58], and the Landau resonance suppression caused by a beam-like electron component may play an important role in generating those oblique waves[47].

The simulation results take some moderate obliquity for granted, regardless of its origin, based on near-equatorial observations. As they are based on a uniform background field they indicate that magnetic field inhomogeneity is not the key condition for the generation of banded chorus waves, in contrast to the mechanism proposed by previous studies[59]. However, the geomagnetic dipole field provides mirror forces which reflect Landau accelerated electrons back to the source region and suppress the chorus excitation at 0.5 $f_{ce}$. Moreover, in the geomagnetic dipole field chorus waves become oblique propagating to higher latitudes, and the Landau resonant phase velocity also increases. Therefore, electron parallel acceleration occurs not only at the equatorial Landau resonant energy but also at energies above it, leading to an enhanced separation of the two anisotropic electron components and consequently a more pronounced spectral gap. The enhanced parallel electron distribution in the simulation suggests electron loss to the atmosphere, and is consistent with the electron precipitation recorded by spacecraft at low altitudes of roughly the same magnetic line as the satellite (Supplementary Discussion 2 and Supplementary Fig. 4).

Figure 5 schematically illustrates the generation mechanism of two-band chorus waves deduced from a combination of Van Allen Probes observations and self-consistent particle-in-cell simulations: (1) The initial anisotropic electron distribution generates a single chorus wave band over a broad frequency range (also see Supplementary Fig. 6). (2) These one-band chorus waves immediately accelerate electrons via Landau resonance near the equator and at higher latitudes, splitting the anisotropic electrons into two energy ranges, separated by a more isotropic range. (3) The new chorus waves are excited at the upper and lower bands by two separate anisotropic electron populations. The spectral gap is formed at 0.5 $f_{ce}$, because the Landau resonant energies of chorus waves over a broad frequency range are close to the cyclotron resonant energies of chorus waves at 0.5 $f_{ce}$. The separation of anisotropic electrons into two energy ranges, caused by Landau resonant parallel acceleration, is the key process here. As it occurs very quickly, the two-band chorus wave structure is common in the outer radiation belt, while chorus waves with spectra across 0.5 $f_{ce}$, which according to our model should only occur at the initial excitation stage, are relatively uncommon[60–62]. This study shows the complexity of the wave-particle interactions, that is, the initially excited waves cause immediate feedback to the plasma environment and significantly modify the particle distribution, which subsequently generates a new wave structure.

Chorus waves, ubiquitous across planetary environments and plasma conditions, have a characteristic two-band structure whose understanding had not been previously fully understood. These waves play a critical role in accelerating seed electrons to relativistic

energies, contributing to the energetic electron losses, the diffuse and pulsating planetary aurorae occurances[8–12,22], and global energy dissipation at Earth[63,64] and other planets. Therefore, understanding their characteristics and evolution is crucial for deciphering how planetary environments operate and for advancing space weather modeling at Earth. Our theory, supported by observations and simulations, provide a natural and simple explanation to this phenomenon under most conditions, and make predictions that can be further tested with specialized observational studies.

## Methods

**Electron PSD distribution fitting**. We use a combination of bi-Maxwellian distribution and kappa distribution to fit the measured electron PSD shown in Fig. 2a. These two distributions are expressed as

$$f = \frac{n_o}{\pi^{3/2} v_{T\perp}^2 \, v_{T\parallel}} \exp\left( -\frac{v_\perp^2}{v_{T\perp}^2} - \frac{\left(v_\parallel - v_b\right)^2}{v_{T\parallel}^2} \right), \tag{1}$$

$$f = \frac{n_0 2^{2\kappa-1}(\kappa - 1/2)\Gamma^2(\kappa)}{\pi^2 \sqrt{\kappa}\,\Gamma(2\kappa) v_{T\perp}^2 \, v_{T\parallel}} \left( 1 + \frac{v_\perp^2}{\kappa v_{T\perp}^2} + \frac{v_\parallel^2}{\kappa v_{T\parallel}^2} \right)^{-\kappa-1}. \tag{2}$$

Here $\left(v_\perp, v_\parallel\right)$ are the particle velocities in the transverse and parallel directions with respect to the background magnetic field, and $\left(v_{T\perp}, v_{T\parallel}\right)$ are the transverse and parallel thermal velocities. The beam velocity is represented by $v_b$, $n_0$ is the number density, and $\Gamma(\kappa)$ is the Gamma function. The $\kappa$ index is the exponent of the PSD at high energies, and the kappa distribution (Eq. 2) reduces to a bi-Maxwellian distribution in the $\kappa \to \infty$ limit.

In the present case study, the plasma density inferred from the $f_{UH}$ and magnetic field measurements is $8.0\,\mathrm{cm}^{-3}$. Since the density is mainly contributed by the cold, bi-directional component, this density measured at the spacecraft location (Latitude $= -6.0°$) represents the density at equator. We fit the measured electron PSD with 5 electron components using the trial and error method. The fitting parameters, shown in Table 1, are chosen carefully so that the numerical results represent the measured electron distribution and the total density also matches the measured one. Since the electrons had already been relaxed by chorus waves, we use a larger anisotropy than measured to model the anisotropic electrons. (see Supplementary Fig. 5 for a comparison).

**Electron velocity distribution setting in the particle-in-cell simulation**. While a Gaussian distribution sample can be generated by standard procedures such as the Box–Muller method[65], there is currently no standard method to generate a kappa distribution sample. In the present study, the initial electron velocity distribution is set based on the distribution probability in each $\left(v_{\perp i}, v_{\perp i+1}, v_{\parallel j}, v_{\parallel j+1}\right)$ cell. Here, both $v_{\perp i}$ and $v_{\parallel j}$ are geometric sequences with 90 elements from $5 \times 10^{-4}$ to 0.99 in units of light speed. The number of particles in the $(i, j)$ cell is an integer nearest to

$$N \int_{v_{\perp i}}^{v_{\perp i+1}} \int_{v_{\parallel j}}^{v_{\parallel j+1}} f\left(v_\perp, v_\parallel\right) 2\pi v_\perp \mathrm{d}v_\perp \mathrm{d}v_\parallel, \tag{3}$$

where N~8 million is the total electron number. The electron velocity is randomly distributed within each velocity cell.

**Estimation of Landau resonant velocity**. The Landau resonant velocities in this study is calculated based on cold plasma theory[47]. The refractive index $n$ of a parallel propagating chorus wave is

$$n^2 \simeq 1 + \frac{\omega_{pe}^2}{\omega(\omega_{ce} - \omega)}. \tag{4}$$

For $\omega = 0.5\,\omega_{ce}$, the Landau resonant velocity is

$$v_{Landau} = \frac{\omega}{k_\parallel} = \frac{c}{n} \simeq c/\sqrt{1 + 4\omega_{pe}^2/\omega_{ce}^2}. \tag{5}$$

In the cases of this study, the $\omega_{pe}/\omega_{ce}$ ratio is generally larger than 3, therefore $v_{Landau} \simeq c\omega_{ce}/2\omega_{pe} = 0.5\,v_{Ae}$ where $v_{Ae}$ is the electron Alfvenic velocity, and the representative resonant energy is $E_{Landau} = mv_{Ae}^2/8$. The Landau resonant velocities for quasi-parallel propagating banded chorus waves are typically within a narrow range around the representative velocity (Fig. 2b).

## Data availability
The Van Allen Probe data are publicly available from http://www.RBSP-ect.lanl.gov/ and https://emfisis.physics.uiowa.edu/. All data supporting the findings of this study will be available upon request.

## Code availability
The Van Allen Probe data analysis is carried out using the publicly available SPEDAS software (version 3.1, http://spedas.org). The 1-2/2D Darwin particle-in-cell simulation code are publicly available on https://picksc.idre.ucla.edu/software/skeleton-code/software-skeleton-code-serial/.

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

## Acknowledgements

We thank UCLA Particle-in-Cell and Kinetic Simulation Software Center for providing the simulation codes. J.L. and J.B. acknowledge the grant DE-SC0010578 and NASA grants NNX16AG21G, NNX14AN85G, HTIDES NNX16AG21G, HGCR NNX15AI96G. J.L., J.B., and X.A. acknowledge the NSF grant AGS-1923126. R.M.T. and J.L. acknowledge NASA grant NNX14AI18G. WL acknowledges the NASA grant NNX17AD15G, AFOSR grant FA9550-15-1-0158, NSF grant AGS-1723588, and the Alfred P. Sloan Research Fellowship FG-2018-10936. This work was supported by RBSP-ECT and EMFISIS funding provided by JHU/APL contract No. 967399 and 921647 under NASA's prime contract No. NAS5-01072. This paper is dedicated to the memory of the late Richard Thorne, a pioneer of wave-particle interaction research, who passed on July 12th 2019 during the consideration of this manuscript.

## Author contributions

J.L., J.B., and X.A. conceived the idea and oversaw the project. J.L. carried out the data survey and numerical simulation. X.A. developed the simulation methodology. W.L. assisted with linear interaction theory and data analysis. R.M.T., C.T.R., V.A., and B.N. assisted with theoretical analysis and result interpretation. X.S. analyzed the low altitude observations. W.S.K., G.B.H., and D.P.H. provided the EMFISIS data. H.O.F. provided the HOPE data. D.N.B. and H.E.S led the RBSP-ECT energetic particle team. All authors commented on the paper.

## Competing interests

The authors declare no competing interests.
