## [Peer Review File · Nature Communications]

Reviewers' Comments:

Reviewer #1:

Remarks to the Author:

Summary

This manuscript presents the generation mechanism for two-banded chorus observed in the Earth's radiation belts using comprehensive satellite measurements of waves and electrons, together with a simulation study. The two-band chorus is commonly observed at Earth and other magnetized planets such as Jupiter and Saturn, while the generation mechanism has not been identified well for decades despite numerous theoretical and observational studies. Chorus waves play crucial roles not only the dynamics of the outer radiation belt (such as acceleration and loss processes of high energy electrons) but also atmospheric physics, since the waves cause energetic particle precipitations that may affect the temperature, density, and composition of atmospheric molecules. Thus this study potentially has an impact on the research area.

The authors find signatures of the electron acceleration via Landau resonance with chorus waves and argue that the electron acceleration results in a split of a single anisotropic distribution into low and high energy anisotropic components. They suggest that the two anisotropic distributions are responsible for the generation of lower- and upper-band chorus. This suggestion is tested by the simulation study, and they reproduce the electron acceleration and the formation of two-banded chorus observed by the satellite. Although the signature of the electron Landau resonance was reported by the satellite measurements [Min et al., 2014], the authors show more convincing observational results and strengthen their argument combining the excellent simulation study with the observation.

The reviewer believes that the approach combining the satellite measurements with the simulation study is novel to prove their generation model. Their observational data and the simulation result are sound and well described, with some exceptions that the reviewer details below. The necessary information to reproduce the results is available in the manuscript.

In summary, the reviewer believes that the manuscript has the potential to meet the standard for publication on Nature Communications as the presented results may have an impact on the research field. On the other hand, the reviewer thinks that the manuscript needs some clarification with respect to some issues shown in the following.

Specific comments

Line 57:

Referencing seems to be inappropriate. Why is the paper on hiss waves referred here?

Line 69-72:

The reviewer thinks that the appropriate location of this sentence seems to be at the top of the next paragraph.

Line 82-86:

As the authors indicate in the manuscript, the linear growth rate calculation in Figure 2 fails to explain the observed wave frequency spectrogram. For example, the observation shows upper-band chorus lies in the frequency range from 0.5 to 0.7 fce, while the linear theory predicts that the wave growth is possible in the frequency range of 0.7-0.8 fce. This result does not support the authors' conclusion that two anisotropic distributions are responsible for excitation of banded chorus waves.

Line 117-121:

The authors argue that the distributions shown in Figure 3 demonstrate the result of Landau

acceleration by chorus waves. However, in some cases, it seems to the reviewer that the distributions are not the result of the Landau acceleration (e.g., 2013-09-24, 2014-02-28, 2014-04-19, 2014-04-21, 2014-05-09, 2014-08-01), since a kink (or a plateau) appears near the resonance energy for all pitch angles. Does the mechanism considered by the authors explain the feature? Also, It is not clear which events clearly show $f(18\text{deg}) > f(36\text{deg})$.

Figure1:

Figure 1b in the manuscript and that attached to the manuscript seem to be different. Which is the correct version?

Figure 5:

The reviewer believes that the authors need to redraw the schematic illustration shown in Figure 5 (especially Figure 5b) after the consideration described below, since the illustration may not be correctly understood by experts who have studied chorus waves.

Many previous works have indicated that discrete rising tone chorus elements seen in Figure 1c are excited through nonlinear wave-particle interactions between electrons and whistler mode waves predicted by the linear theory. (I should note that importance of the nonlinear aspect on chorus excitation is not described in the current manuscript at all, while the reviewer understands that the aspect is not the critical point of this manuscript.) Based on the theory of chorus excitation, chorus elements are generated after the excitation of linear phase whistler mode waves. It is essential to consider the time scale for Landau damping/acceleration and triggering of chorus elements to show the schematic illustration. If the damping/acceleration process is faster than chorus excitation, it is quite difficult for chorus elements to go through 0.5 fce. If chorus excitation is faster than the damping process, one chorus element is split into lower- and upper-band chorus waves. However, it is quite difficult to explain the lack of one-to-one correspondence between lower- and upper-band chorus waves which is seen in Figure 1c.

If the Figures 5b and 5d show wave power distributions as a function of frequency, the illustrations are acceptable for the experts.

Reference:

Min, K., K. Liu, W. Li, Signatures of electron Landau resonant interactions with chorus waves from THEMIS observations, *J. Geophys. Res. Space Physics*, 119, 5551–5560 (2014), doi: 10.1002/2014JA019903.

Reviewer #2:

Remarks to the Author:

ELF/VLF chorus emissions are intense electromagnetic plasma waves that are naturally and spontaneously excited near the magnetic equatorial plane outside the plasmasphere during periods of magnetic disturbance. These emissions are believed to play an important role in the acceleration of keV radiation belt electrons to MeV energies during the disturbed time periods. Chorus emissions frequently occur in two frequency bands with a gap near half gyrofrequency. Despite their common occurrence and importance in electron acceleration, generation mechanism of chorus including the double band (two-band) structure remains a subject of ongoing research.

The manuscript (MS) under review has provided simultaneous measurements of two-band chorus and electron phase space density (PSD) illustrating a close correlation between the double band structure and two distinct anisotropic components in electron PSD at few hundred eV and a few keV separated by an isotropic PSD at intermediate energies. Using 1-D particle in cell (PIC) simulations, authors have self-consistently produced observed double band chorus and electron PSD with two distinct anisotropic PSD components.

The proposed chorus generation mechanism is interesting because it self-consistently explains both the observations of the two-band chorus and two distinct components in the anisotropic PSD.

Though the MS has provided 20 good cases of the simultaneous observations of two-band chorus and PSD with two distinct anisotropic populations, neither the data presented in this paper as nor the generation mechanism provided are not entirely novel. For example, Schriver et al. [2010], Liu et al. [2011], Fu et al. [2014], Ratcliffe and Watt [2017] have proposed similar generation mechanisms for two-band chorus. Schriver et al. [2010] and Fu et al. [2014] have provided similar wave and particle data from Cluster 4 and Van Allen Probes satellites, respectively. The MS does not adequately discuss how the mechanisms proposed and the data presented are fundamentally different from those presented in the literature. The MS has not referenced Schriver et al. [2010], one of the first papers to consider chorus generation mechanism similar to one proposed in this MS.

One of my major concern is that the proposed mechanism by authors to explain the often observed double-banded structure implies that there be a strong correlation between the lower and upper band chorus. However, there are several previously noted features of the double-banded chorus that are not consistent with this implication. For example, Burtis and Helliwell [1975] have noted that when two bands are present simultaneously the individual elements rarely show a one-to-one correlation between bands. That such a correlation is lacking can also be seen from Figure 1 of the MS.

The proposed generation mechanism also needs to explain other observed features of the chorus spectrum [e.g., Burtis and Helliwell, 1975] and wave normal direction [e.g., Li et al., 2016; Hartley et al., 2019] as noted in the literature. Quasi-parallel Chorus wave normal angles used in this study for simulations may not be a common feature of either lower or upper band chorus as suggested by recent wave normal analysis [Li et al., 2016]. The past research has shown that inhomogeneity in the geomagnetic field is essential in explaining the discrete rising/falling structure of chorus elements [e.g., Omura et al. 2009]. Authors should explain how their simulations without using inhomogeneity in the magnetic field can produce rising emissions.

The details of the methods used are insufficient, and hence the results presented may be difficult to reproduce. See below.

a) Applicability of Darwin code for chorus simulations: In the past, mostly full electromagnetic (EM) particle in cell (PIC) codes were used for chorus simulations [e.g., Fu et al., 2014]. Darwin's code neglects the transverse component of displacement current. Why was Darwin's code chosen over full EM PIC codes? Also, Darwin's code permits relaxed integration in time which may lead to numerical instability. What is the time step used for the simulations presented? Will simulation results be different if a full EM PIC code is used?

b) Insufficient methodological detail

Examples: Lines 208-211 It is not clear how the authors concluded to fit electron phase space density (PSD) distributions measured by the HOPE instrument as a summation of bi-Maxwellian distribution and Kappa (power law) distribution. Is such fitting done previously? If so references must be provided. What is the methodology used for fitting? Was it trial and error? How were the input parameters chosen and varied? What are the reasonable ranges of parameters chosen? Is n_0 same as N_e in Table 2? Is it N_e at the equator?

c) Line 217: It is not clear what authors mean by 5 electron components to fit. Also, from observations, electron density was found to be roughly constant (line 52) whereas from table 2 it can be noted that N_e varied in the range 7.2 to 0.001. How do the authors justify such large variation in N_e to fit the data?

d) Lines: 136-139: What is the rationale for describing isotropic cold plasma by the Kappa distribution? Is this kind of a distribution justified? At what location are the inputs prescribed? What is the value of n_0 used in simulations?

e) Lines 221-223: How are the growth rates calculated? What is fitted analytical distribution? Was this method validated in the past? If so provide references.

f) Other relevant questions to simulations and methodology that need to be addressed include: what are the input and outputs of the code/simulation? What is the typical simulation time? Are the chosen inputs reasonable? The validity of the code for the simulations presented in the paper? The rationale for choosing the simulation code?

g) It would be useful to show the dynamic spectra of chorus emissions from simulations during various stages in simulation and compare simulation results with those observed.

h) Missing references: for example, references for bi-Maxwellian distribution, Kappa distribution, Landau resonant energy calculations, Box-Muller method are missed. References for the validity of the models used in the paper are missing.

Minor comments (not extensive):

1. Lines 47-52: It will be useful to give Van Allen Probe orbit information, L-shell, MLT at which chorus was observed, measured f_{uh} , calculated values of f_{pe} and f_{ce} .
2. Line 53, Are the f_{pe} and f_{ce} values along the orbit roughly constant?
3. The values of f_{pe}/f_{ce} used for calculating resonant energies (Figure 2; $f_{pe}/f_{ce}=5.5$) and particle in cell simulations ($f_{pe}/f_{ce}=5$) was different why?
4. The discussion section is more like summary and concluding remarks
5. Organization of the paper: It will help the reader if the methods for calculating growth rates and fitting electron distributed is presented earlier in the simulations section rather than at the end of the paper.
6. How do the authors explain the variability of the banded chorus gap width?

References:

Burtis, W. J. and Helliwell, R. A. (1976) Magnetospheric chorus: occurrence patterns and normalized frequency. *Planet. Space Sci.* 24, 1007.

Hartley, D. P., Kletzing, C. A., Chen, L., Horne, R. B., & Santolik, O. (2019). Van Allen Probes observations of chorus wave vector orientations: Implications for the chorus-to-hiss mechanism. *Geophysical Research Letters*, 46, 2337–2346. <https://doi.org/10.1029/2019GL082111>

Li, W., O. Santolik, J. Bortnik, R. M. Thorne, C. A. Kletzing, W. S. Kurth, and G. B. Hospodarsky (2016), New chorus wave properties near the equator from Van Allen Probes wave observations, *Geophys. Res. Lett.*, 43, 4725–4735, doi:10.1002/2016GL068780

Schrifer, D., et al. (2010), Generation of whistler mode emissions in the inner magnetosphere: An event study, *J. Geophys. Res.*, 115, A00F17, doi:10.1029/2009JA014932.

Reviewer #3:

Remarks to the Author:

In this manuscript, the authors at first observed that two-band structure of chorus waves are usually associated with two anisotropic electron components, and then they tried to use 1-D PIC simulations to demonstrate that the initially excited single-band chorus waves can alter electron distribution via Landau resonance and divides the electron anisotropy into a low and high energy components, and at last both the upper- and low-band chorus waves are excited. The idea is

attractive, however, I think that the evidence is not enough to support the conclusion. The reasons are described as follows.

1. After I read the manuscript, I am a little confused on where the two-band structure of chorus waves is formed. From the Figure 1, the waves have obvious rising tone, so they have already left the source region of the excited waves. However, in the simulation, the authors are describing the excitation of whistler waves by an electron temperature anisotropy, it is in the source region of the excited waves.

2. In the observations, the chorus waves have a two-band structure, and the amplitude of the lower band is larger than that of the upper band. However, from Figure 2, the growth rate of the upper band is larger than that of the lower band. If the two bands of the chorus waves are excited separately by a low and high energy electron components, it is difficult to explain it. Again, if according to the electron distribution the whistler waves have a high growth rate, they should be in the source region. An electron temperature anisotropy should excite whistler waves propagating both parallel and antiparallel to the magnetic field, then how can the waves have an obvious rising tone.

3. In simulations of whistler waves excited by an electron temperature anisotropy, during the relaxation of the temperature anisotropy, the frequency will drift to a value. It is not strange to observe a gap due to Landau resonance if the excited waves obliquely propagate. So, it is not clear whether the low-band of whistler waves are excited by the high energy component.

Responses to review comments

The electron distribution data measured by the HOPE instrument has been updated in March 2019. In response, we updated the figure, and changed the fitting parameters accordingly in the new manuscript. The updated electron distribution basically shows the same characteristics as before. One major change is that the overall fluxes are doubled compared to the previous version, but this change does not change our conclusion.

Reply to comments by Reviewer #1

Reviewer #1 (Remarks to the Author):

Summary

This manuscript presents the generation mechanism for two-banded chorus observed in the Earth's radiation belts using comprehensive satellite measurements of waves and electrons, together with a simulation study. The two-band chorus is commonly observed at Earth and other magnetized planets such as Jupiter and Saturn, while the generation mechanism has not been identified well for decades despite numerous theoretical and observational studies. Chorus waves play crucial roles not only the dynamics of the outer radiation belt (such as acceleration and loss processes of high energy electrons) but also atmospheric physics, since the waves cause energetic particle precipitations that may affect the temperature, density, and composition of atmospheric molecules. Thus, this study potentially has an impact on the research area.

The authors find signatures of the electron acceleration via Landau resonance with chorus waves and argue that the electron acceleration results in a split of a single anisotropic distribution into low and high energy anisotropic components. They suggest that the two anisotropic distributions are responsible for the generation of lower- and upper-band chorus. This suggestion is tested by the simulation study, and they reproduce the electron acceleration and the formation of two-banded chorus observed by the satellite. Although the signature of the electron Landau resonance was reported by the satellite measurements [Min et al., 2014], the authors show more convincing observational results and strengthen their argument combining the excellent simulation study with the observation.

The reviewer believes that the approach combining the satellite measurements with the simulation study is novel to prove their generation model. Their observational data and the simulation result are sound and well described, with some exceptions that the reviewer details below. The necessary information to reproduce the results is available in the manuscript.

In summary, the reviewer believes that the manuscript has the potential to meet the standard for publication on Nature Communications as the presented results may have an impact on the research field. On the other hand, the reviewer thinks that the manuscript needs some clarification with respect to some issues shown in the following.

Specific comments

Line 57:

Referencing seems to be inappropriate. Why is the paper on hiss waves referred here?

Reply: Thanks for pointing out this mistake. We deleted this citation number in the text.

Line 69-72:

The reviewer thinks that the appropriate location of this sentence seems to be at the top of the next paragraph.

Reply: Thanks for this suggestion. In the revised manuscript, we relocated this sentence to the next paragraph.

Line 82-86:

As the authors indicate in the manuscript, the linear growth rate calculation in Figure 2 fails to explain the observed wave frequency spectrogram. For example, the observation shows upper-band chorus lies in the frequency range from 0.5 to 0.7 f_{ce} , while the linear theory predicts that the wave growth is possible in the frequency range of 0.7-0.8 f_{ce} . This result does not support the authors' conclusion that two anisotropic distributions are responsible for excitation of banded chorus waves.

Reply: The linear growth calculation shown in the original Figure 2 is successful in explaining the main spectral feature of the chorus waves, i.e., the two-band structure separated by a gap around 0.5 f_{ce} .

The electron fluxes measured by the HOPE instrument has been updated in March 2019. We thus updated Figure 2a (see figure below), which basically presents the same feature as the previous one. One major change is that the overall electron flux, thus the phase space density, doubled. We updated the electron PSD fitting parameters as seen in Table 2, thus the newly calculated linear growth rates, shown in Figure 2c below, demonstrates wave growth at 0.6-0.7 f_{ce} , roughly consistent with the observed upper-band chorus.

Figure R1. The measured electron PSD and the calculation of linear growth rates. This figure is the revised Figure 2 in the manuscript.

Line 117-121:

The authors argue that the distributions shown in Figure 3 demonstrate the result of Landau acceleration by chorus waves. However, in some cases, it seems to the reviewer that the distributions are not the result of the Landau acceleration (e.g., 2013-09-24, 2014-02-28, 2014-04-19, 2014-04-21, 2014-05-09, 2014-08-01), since a kink (or a plateau) appears near the

resonance energy for all pitch angles. Does the mechanism considered by the authors explain the feature? Also, It is not clear which events clearly show $f(18\text{deg}) > f(36\text{deg})$.

Reply: Although the kink in the electron distribution at $\sim 1-10$ keV is also observed at large pitch angles in several events, the acceleration at small pitch angles is generally more profound. One possible mechanism that causes the kink at large pitch angles could be electron cyclotron resonant acceleration caused by upper-band chorus at higher latitudes. Since more than half of those cases do not show acceleration at all pitch angles, and the present paper aims to address the origin of two-band structure instead of electron acceleration, we do not discuss this issue in the current manuscript. The electron acceleration and the cause of a kink at $\sim 90^\circ$ pitch angle associated with two-band chorus, or only upper-band or lower-band chorus will be explored in our future project.

Most of those events show $f(18 \text{ deg}) > f(36 \text{ deg})$ at Landau resonant energies. The 2014-04-13, 2014-05-04, 2014-06-03 events do not show the feature of $f(18 \text{ deg}) > f(36 \text{ deg})$, however, they show evident PSD plateau at Landau resonant energies. The only two exceptions that do not show either $f(18 \text{ deg}) > f(36 \text{ deg})$ or PSD plateau are the 2014-04-19 and 2014-04-21 events. We revised the sentence on Lines 128-130:

“The two exceptions are the 04/19/2014 and 04/21/2014 events, possibly because the Landau resonant acceleration and the formation of two-band chorus waves are still in process.”

Figure1:

Figure 1b in the manuscript and that attached to the manuscript seem to be different. Which is the correct version?

Reply: The independent figure is correct. The Figure previously in the manuscript does not contain the L, LAT and MLT information. We have used the correct one in the revised manuscript.

Figure 5:

The reviewer believes that the authors need to redraw the schematic illustration shown in Figure 5 (especially Figure 5b) after the consideration described below, since the illustration may not be correctly understood by experts who have studied chorus waves.

Many previous works have indicated that discrete rising tone chorus elements seen in Figure 1c are excited through nonlinear wave-particle interactions between electrons and whistler mode waves predicted by the linear theory. (I should note that importance of the nonlinear aspect on chorus excitation is not described in the current manuscript at all, while the reviewer understands that the aspect is not the critical point of this manuscript.) Based on the theory of chorus excitation, chorus elements are generated after the excitation of linear phase whistler mode

waves. It is essential to consider the time scale for Landau damping/acceleration and triggering of chorus elements to show the schematic illustration. If the damping/acceleration process is faster than chorus excitation, it is quite difficult for chorus elements to go through $0.5 f_{ce}$. If chorus excitation is faster than the damping process, one chorus element is split into lower- and upper-band chorus waves. However, it is quite difficult to explain the lack of one-to-one correspondence between lower- and upper-band chorus waves which is seen in Figure 1c.

If the Figures 5b and 5d show wave power distributions as a function of frequency, the illustrations are acceptable for the experts.

Figure R2. The new schematic illustration (Figure 5 in the manuscript).

Reply: This figure is redrawn as shown above. We use the same event studied in this paper to demonstrate the one-band and two-band chorus waves with wave spectral power information. As pointed out by the reviewer, this paper does not address the mechanism of rising-tone structures, therefore, we chose not to illustrate the rising-tone structures. The one-band chorus waves were measured at 05:25-05:35 UT at $L \sim 4.5$, and the two-band chorus waves are observed at large L -shells. The measured electron PSDs of these two events are shown in Figure S6 in the supplemental material (shown below).

Figure S6. The measured electron PSD distributions associated with one-band and two-band chorus waves shown in Figure 5. The electron PSD distributions associated with one-band chorus waves exhibit one anisotropic component, while those associated with two-band chorus exhibit two anisotropic components.

We would like to point out that the two-band chorus waves in the magnetosphere are not developed from the one-band chorus waves. The initially generated one-band chorus waves could be damped at higher latitudes, or they could propagate to other L-shells and MLTs. The two-band chorus waves are newly generated from the two anisotropic components. In the manuscript, we stated on Lines 209-210 that “...chorus waves with spectra across $0.5 f_{ce}$, which according to our model should only occur at the initial excitation stage, are rare.” To further clarify the point that the two-band chorus waves are newly generated in the source region, instead of being developed from the one-band chorus, we revised the sentence on Lines 203-204 as:

“3) The new chorus waves are excited at the upper and lower bands by two separate anisotropic electron populations.”

Reference:

Min, K., K. Liu, W. Li, Signatures of electron Landau resonant interactions with chorus waves from THEMIS observations, *J. Geophys. Res. Space Physics*, 119, 5551–5560 (2014), doi:10.1002/2014JA019903.

Reply: This paper was cited in the original manuscript. See Lines 106-107 in the new manuscript.

“This is similar to electron PSD plateaus created through Landau resonant acceleration by chorus waves, mostly occurring in the lower-band^{51,52}.”

Reply to comments by Reviewer #2

Reviewer #2 (Remarks to the Author):

ELF/VLF chorus emissions are intense electromagnetic plasma waves that are naturally and spontaneously excited near the magnetic equatorial plane outside the plasmasphere during periods of magnetic disturbance. These emissions are believed to play an important role in the acceleration of keV radiation belt electrons to MeV energies during the disturbed time periods. Chorus emissions frequently occur in two frequency bands with a gap near half gyrofrequency. Despite their common occurrence and importance in electron acceleration, generation mechanism of chorus including the double band (two-band) structure remains a subject of ongoing research.

The manuscript (MS) under review has provided simultaneous measurements of two-band chorus and electron phase space density (PSD) illustrating a close correlation between the double band structure and two distinct anisotropic components in electron PSD at few hundred eV and a few keV separated by an isotropic PSD at intermediate energies. Using 1-D particle in cell (PIC) simulations, authors have self-consistently produced observed double band chorus and electron PSD with two distinct anisotropic PSD components.

The proposed chorus generation mechanism is interesting because it self-consistently explains both the observations of the two-band chorus and two distinct components in the anisotropic PSD. Though the MS has provided 20 good cases of the simultaneous observations of two-band chorus and PSD with two distinct anisotropic populations, neither the data presented in this paper as nor the generation mechanism provided are not entirely novel. For example, Schriver et al. [2010], Liu et al. [2011], Fu et al. [2014], Ratcliffe and Watt [2017] have proposed similar generation mechanisms for two-band chorus. Schriver et al. [2010] and Fu et al. [2014] have provided similar wave and particle data from Cluster 4 and Van Allen Probes satellites, respectively. The MS does not adequately discuss how the mechanisms proposed and the data presented are fundamentally different from those presented in the literature. The MS has not referenced Schriver et al. [2010], one of the first papers to consider chorus generation mechanism similar to one proposed in this MS.

Reply: Thank you for a very thorough reading and insightful comment on our work. We note that Schriver et al. [2010] proposed a different mechanism from the one we present in our paper. That publication showed a chorus event with two-band structure, but the electron distribution only had one anisotropic component. The linear growth rates show that the chorus waves should only grow in the upper band. Schriver et al. explained that the lower band is generated due to a nonlinear wave-wave coupling mechanism. To illustrate our point, we cite the paper by Schriver et al. on Lines 34-36:

“It has been proposed that the lower-band can be triggered by the upper-band [Schriver et al., 2010] (or vice versa [Gao et al., 2017]) via a nonlinear wave-wave coupling, but this theory cannot explain general cases in which the two bands have no correlation.”

Our manuscript stated on Lines 206-208 that “The separation of anisotropic electrons into two energy ranges, caused by Landau resonant parallel acceleration, is the key process here”. Fu et al. [2014] showed only one case study of two-band chorus associated with two anisotropic components, thus we commented on Lines 39-40:

“However, it is unknown whether this causality is general or just a coincidence in one specific case, and how the subtle multi-component electron distribution is developed.”

Figure R3. The initial electron PSD for 5 PIC simulations presented by Ratcliffe and Watt [2017]. Ratcliffe and Watt [2017] used 3 bi-Maxwellian distributions (a cold, a warm and a hot components) to model the initial electron distributions, as did in the study by Fu et al. [2010]. Figure R3 shows the initial electron PSD for the 5 experiments presented in Ratcliffe and Watt [2017]. The representative Landau resonant velocities ($v_{\text{Landau}} = 0.14c$) in those cases are indicated by cyan arrows. We clearly see two anisotropic components in all those simulations, and the separation energy is roughly around the Landau resonant energies. Two simulation

experiments (M1 and M3) produced a spectral gap at ~ 0.5 fce. The simulation results of M2 is ambiguous, since the figure shows a spectral gap, but the authors conclude that “one of our simulations, M2, did not show a convincing gap feature, as its upper band is mostly absent”. The M4 and M5 simulations produced a spectral gap at 0.2 fce instead of at 0.5 fce. Again, this paper did not address how the subtle two anisotropic components are developed. We revised the text on Lines 40-42 as:

“A recent particle-in-cell simulation³³ demonstrated growth of two-band chorus waves resulting from two anisotropic electron populations, but did not address how the electron population developed into two anisotropic components.”

One of my major concern is that the proposed mechanism by authors to explain the often observed double-banded structure implies that there be a strong correlation between the lower and upper band chorus. However, there are several previously noted features of the double-banded chorus that are not consistent with this implication. For example, Burtis and Helliwell [1975] have noted that when two bands are present simultaneously the individual elements rarely show a one-to-one correlation between bands. That such a correlation is lacking can also be seen from Figure 1 of the MS.

The proposed generation mechanism also needs to explain other observed features of the chorus spectrum [e.g., Burtis and Helliwell, 1975] and wave normal direction [e.g., Li et al., 2016; Hartley et al., 2019] as noted in the literature. Quasi-parallel Chorus wave normal angles used in this study for simulations may not be a common feature of either lower or upper band chorus as suggested by recent wave normal analysis [Li et al., 2016]. The past research has shown that inhomogeneity in the geomagnetic field is essential in explaining the discrete rising/falling structure of chorus elements [e.g., Omura et al. 2009]. Authors should explain how their simulations without using inhomogeneity in the magnetic field can produce rising emissions.

The details of the methods used are insufficient, and hence the results presented may be difficult to reproduce. See below.

Reply: The upper-band and the lower-band chorus emission do not necessarily need to have a one-to-one correlation in the data (Figure 1) and in our simulation. (According to our preliminary survey, a small portion of the Van Allen Probe observations show one-to-one correlation between the two bands, while most cases do not show a strong correlation. This will be another study.) The simulated wave spectra in Figure 4b, 4d and 4f are shown as a function of frequency (ω) and k mode in order to demonstrate both the dispersion relation and the spectral intensity of those waves, thus, they do not represent the rising-tone structure seen in typical frequency-time spectrograms. We illustrated, inside the frame of Figure 4b, that the k mode number indicate the number of wavelengths in the simulation space. We changed the title of the subfigures 4b, 4d and 4f to “FFT over xx-xx T_{gyro} ”, and changed the x-labels to “ k mode number”. We added the following sentence on Lines 160-162:

“Note that the wave spectra are shown as a function of frequency (ω) and k mode number (the number of wavelengths in the simulation space), and do not represent the rising-tone structures commonly observed in frequency-time spectrograms.”

Since we are not focusing on simulating the generation of rising tones, the inhomogeneity of the magnetic field is not critical in our simulation. The mechanism of the ring-tone characteristics is one major problem in nonlinear wave-particle interactions in space plasmas, and is beyond the scope of this study. We also changed the schematic figure (See Figure 5 in manuscript) which does not show the rising-tone feature.

Although this paper studies the generation of two-band chorus waves observed with small wave normal angles (say, smaller than 30°), the basic idea that the initially generated chorus can separate the anisotropic electron populations into two components may also apply to the generation of banded chorus waves with large normal angles. However, for very oblique chorus waves with normal angles close to resonance cone angle, while the electron anisotropy is still the source, the Landau resonance suppression also plays an important role. We added the following sentence on Lines 185-188:

“While most chorus waves have small wave normal angles⁵⁸, banded chorus waves with a quasi-electrostatic, very oblique lower band is also observed⁵⁹, and the Landau resonance suppression caused by a beam-like electron component may play an important role in the chorus wave generation process⁴⁷.”

a) Applicability of Darwin code for chorus simulations: In the past, mostly full electromagnetic (EM) particle in cell (PIC) codes were used for chorus simulations [e.g., Fu et al., 2014]. Darwin’s code neglects the transverse component of displacement current. Why was Darwin’s code chosen over full EM PIC codes? Also, Darwin’s code permits relaxed integration in time which may lead to numerical instability. What is the time step used for the simulations presented? Will simulation results be different if a full EM PIC code is used?

Reply: The Darwin code neglects the transverse displacement current and thus excludes light waves, but leaves the physics of whistler waves unaffected which is our main concern in this paper. The full EM PIC code has a constraint on the time step which is set by the Courant condition

$$\Delta t < \frac{v_{te}}{c\omega_{pe}}$$

where v_{te} is the electron thermal velocity. The Darwin code removes such a constraint, therefore a large time-step can be used, which greatly improve the computational efficiency. In our study, $\frac{v_{te}}{c} = 0.02$. We used a time step of $0.2 \omega_{pe}^{-1}$. We added the following words on Lines 138-142in the revised manuscript:

“We use the Darwin code, which neglects the transverse displacement current but leaves the physics of whistler mode waves unaffected. It removes the time-step constraint of the full electromagnetic code, and the computational efficiency can be greatly improved by using a large time-step. The Darwin code has been successful in explaining the whistler mode wave excitation and the resultant electron distributions both in space and laboratory plasma environments^{55,56}.”

We added the following sentence on Lines 156:

“The time-step is $0.2 \omega_{pe}^{-1}$.”

We believe that the full EM code, which necessarily needs to run with a much smaller time-step, can produce a similar result as the results presented in this paper.

References:

An, X., et al. (2017a) Electrostatic and whistler instabilities excited by an electron beam, *Phys. Plasmas*, <http://dx.doi.org/10.1063/1.4986511>

An, X., et al. (2017b), On the parameter dependence of the whistler anisotropy instability, *J. Geophys. Res. Space Physics*, 122, 2001–2009, doi:10.1002/2017JA023895.

b) Insufficient methodological detail

Examples: Lines 208-211 It is not clear how the authors concluded to fit electron phase space density (PSD) distributions measured by the HOPE instrument as a summation of bi-Maxwellian distribution and Kappa (power law) distribution. Is such fitting done previously? If so references must be provided. What is the methodology used for fitting? Was it trial and error? How were the input parameters chosen and varied? What are the reasonable ranges of parameters chosen? Is n_0 same as N_e in Table 2? Is it N_e at the equator?

Reply: This fitting has previously been done in a number of studies, such as:

Li, W., et al. (2016), Unraveling the excitation mechanisms of highly oblique lower band chorus waves, *Geophys. Res. Lett.*, 43, 8867– 8875, doi:10.1002/2016GL070386

Li, J., et a. (2017). Chorus wave modulation of Langmuir waves in the radiation belts. *Geophysical Research Letters*, 44, 11,713– 11,721. <https://doi.org/10.1002/2017GL075877>

Li, J., et al. (2018). Local excitation of whistler mode waves and associated Langmuir waves at dayside reconnection regions. *Geophysical Research Letters*, 45, 8793– 8802. <https://doi.org/10.1029/2018GL078287>

The plasma density inferred from the f_{UHR} and magnetic field measurements is 8.0 cm^{-3} . Since the density is mainly contributed by the cold, bi-directional component, the density measured at the spacecraft location (Latitude= 6.2°) approximately represents the density at equator. The input parameters are chosen to be consistent with the measured electron distribution at two pitch

angles: $\alpha_{\text{eq}} = 17.5^\circ$ ($\alpha_{\text{local}} = 18^\circ$) and $\alpha_{\text{eq}} = 75^\circ$ ($\alpha_{\text{local}} = 90^\circ$). We fit different electron components following these procedures:

- 1) Using a Gaussian distribution to fit the cold, bi-directional electrons and the density number (see Figure R4a).
- 2) Adding a Kappa distribution to model the low-energy anisotropic component (Figure R4b). Since the electrons had already been relaxed by chorus waves, a slightly larger anisotropy is used to model the anisotropic electrons before the wave excitation.
- 3) Adding a Kappa distribution to model the high-energy anisotropic component (Figure R4c).
- 4) Adding a Gaussian distribution with a beam population, and an additional kappa distribution to fit the electron PSD at medium energies (Figure R4d).

Figure R4. The electron PSD fitting procedure.

We go through these procedures a few times using the trial and error method until the fitted PSD is close to the measurements. N_0 (8.0 cm^{-3}) is the density at the equator, and it is the same as the total Ne in Table 1 (The previous Table 2 become Table 1 since we have been suggested to describe the fitting method in the text). According to an empirical model [Denton *et al.*, 2004], the density varies with latitude as $n \propto 1/\cos^\alpha \lambda$ where $\alpha = 1.6 - 4.2$, thus, the density at a

latitude of 6.2° is 0.98-0.99 times of that at equator. We revised the “Electron PSD distribution fitting” section to add more details, as seen below.

“Electron PSD distribution fitting. We use a combination of bi-Maxwellian distribution and kappa distribution to fit the measured electron PSD shown in Figure 2a. These two distributions are expressed as

$$f = \frac{n_o}{\pi^{3/2} v_{T\perp}^2 v_{T\parallel}} \exp\left(-\frac{v_{\perp}^2}{v_{T\perp}^2} - \frac{(v_{\parallel} - v_b)^2}{v_{T\parallel}^2}\right). \quad (1)$$

$$f = \frac{n_o 2^{2\kappa-1} (\kappa-1/2) \Gamma^2(\kappa)}{\pi^2 \sqrt{\kappa} \Gamma(2\kappa) v_{T\perp}^2 v_{T\parallel}} \left(1 + \frac{v_{\perp}^2}{\kappa v_{T\perp}^2} + \frac{v_{\parallel}^2}{\kappa v_{T\parallel}^2}\right)^{-\kappa-1}. \quad (2)$$

Here $(v_{\perp}, v_{\parallel})$ are the particle velocities in the transverse and parallel directions with respect to the background magnetic field, and $(v_{T\perp}, v_{T\parallel})$ are the transverse and parallel thermal velocities. The beam velocity is represented by v_b , n_o is the number density, and $\Gamma(\kappa)$ is the Gamma function. The κ index is the exponent of the PSD at high energies, and the kappa distribution (Equation 2) reduces to a bi-Maxwellian distribution in the $\kappa \rightarrow \infty$ limit.

The plasma density inferred from the f_{UHR} and magnetic field measurements is 8.0 cm^{-3} . Since the density is mainly contributed by the cold, bi-directional component, this density measured at the spacecraft location (Latitude= 6.2°) represents the density at equator. We fit the measured electron PSD with 5 electron components using the trial and error method. The fitting parameters, shown in Table 1, are chosen carefully so that the numerical results represent the measured electron distribution and the total density also matches the measured one. Since the electrons had already been relaxed by chorus waves, we use a larger anisotropy than measured to model the anisotropic electrons. (see Figure S4 in the supplementary material for a comparison).”

c) Line 217: It is not clear what authors mean by 5 electron components to fit. Also, from observations, electron density was found to be roughly constant (line 52) whereas from table 2 it can be noted that N_e varied in the range 7.8 to 0.001. How do the authors justify such large variation in N_e to fit the data?

Reply: As shown in the Figure 1a and Figure 5 in this letter, there are 5 electron components in the PSD distribution: 1) a cold, bi-directional component; 2) an anisotropic component at ~ 100 -2000 eV; (3) an anisotropic component at energies > 10 keV; 4) a parallel beam at several keV; (5) an additional component is added to better fit the electron distribution at medium energies. We revised the “Electron PSD distribution fitting” section as can be seen in the new manuscript.

Because the electron PSD at energies > 100 eV is roughly power-law, it is very common that the densities of different electron components vary significantly [e.g., W. Li et al., 2016; J. Li et al., 2017; 2018]

d) Lines: 136-139: What is the rationale for describing isotropic cold plasma by the Kappa distribution? Is this kind of a distribution justified? At what location are the inputs prescribed? What is the value of n_0 used in simulations?

Reply: Since the measured electrons at the lowest energies are roughly in a kappa (power-law) distribution (Figure 2a), we use the kappa distribution to fit both the cold and the warm components. If we were to use a Gaussian distribution to fit the cold component, it would be difficult to smoothly connect the cold and the warm components (See typical fitted PSD profile using bi-Maxwellian distributions in Figure 3 of this letter). Since the cold electrons provide a background to support the chorus wave propagation, the excitation of waves should basically be uninfluenced if we use a Gaussian distribution to fit the cold component.

The initial electron distribution parameters for the PIC simulations are now shown in Table 2. The PIC simulation use normalized physical parameters (e.g., $e = 1$, $m_e = 1$, $\omega_{pe} = 1$, $\epsilon_0 = 1$). There is no parameter of density in the PIC simulation. However, the key parameters (e.g., the f_{pe}/f_{ce} ratio) is basically consistent with observations.

e) Lines 221-223: How are the growth rates calculated? What is fitted analytical distribution? Was this method validated in the past? If so provide references.

Reply: Thanks for this comment and helpful advice. The linear growth rates are calculated using the formula 3.9 in the classic paper “Low-frequency whistler mode” [Kennel, 1966]. Our group had successfully used this method in explaining the generation of various whistler waves under different conditions. For instance, the highly oblique chorus waves [W. Li et al., 2016], the chorus waves associated with modulated Langmuir waves [J. Li et al., 2017], and the whistler mode waves in the dayside reconnection region [J. Li et al., 2018]. We revised the entire paragraph as shown below. The analytical fitted distribution is now described on Lines 82-91 in the text.

“To numerically evaluate the wave growth from the anisotropy instability, we perform a linear growth calculation⁴² which describes the initial phase of wave excitation. Following a similar procedure described in previous studies⁴⁷⁻⁴⁹, we first fit the measured electron phase space density (PSD) distributions as a summation of bi-Maxwellian distribution and Kappa (power law) distribution⁵⁰ (See Methods and Table 1). With the analytically fitted distribution function, the linear growth rates of whistler mode chorus are calculated as the summation of contributions caused by the cyclotron resonance and Landau resonance. The overall linear growth rates, shown in Figure 2c, demonstrate that the chorus waves can grow separately at both the upper and lower bands, while the

chorus waves around $0.5 f_{ce}$ cannot be excited due to insufficient anisotropy at energies of several keV.”

References:

Li, W., et al., Unraveling the excitation mechanisms of highly oblique lower band chorus waves, *Geophys. Res. Lett.*, 43, 8867– 8875 (2016), doi:10.1002/2016GL070386.

Li, J., J. Bortnik, X. An, W. Li, R. M. Thorne, M. Zhou, ... H. E. Spence. Chorus wave modulation of Langmuir waves in the radiation belts. *Geophysical Research Letters*, **44**, 11,713–11,721(2017). <https://doi.org/10.1002/2017GL075877>

Li, J., Bortnik, J., An, X., Li, W., Russell, C. T., Zhou, M., et al.. Local excitation of whistler mode waves and associated Langmuir waves at dayside reconnection regions. *Geophysical Research Letters*, 45, 8793– 8802 (2018). <https://doi.org/10.1029/2018GL078287>

f) Other relevant questions to simulations and methodology that need to be addressed include: what are the input and outputs of the code/simulation? What is the typical simulation time? Are the chosen inputs reasonable? The validity of the code for the simulations presented in the paper? The rationale for choosing the simulation code?

Reply: The inputs of the PIC code include: 1) Electron initial distribution; 2) the f_{pe}/f_{ce} ratio (5); 3) the number of electrons (~8 million); (4) the number of grids (1024); (5) the background magnetic field; (6) the wave normal angle; (7) the boundary condition.

The simulation time is ~1000 electron gyro-periods. The CPU processing time is ~2 day.

We chose to use the Darwin PIC code because it is computationally faster than the full EM PIC codes. The Darwin PIC code neglects the transverse displacement current and thus excludes light waves, but leaves the physics of whistler waves unaffected. The Darwin code had been successfully used in simulating the excitation of whistler mode waves both in space and laboratory environments [e.g., An et al., 2017a; 2017b]. The full EM PIC code has a constraint on the time step which is set by the Courant condition

$$\Delta t < \frac{v_{te}}{c\omega_{pe}^{-1}}$$

where v_{te} is the electron thermal velocity. The Darwin codes remove such a constraint and a large time-step can be used, and therefore, the computational efficiency is greatly improved.

We described the Darwin code in the text on Lines 138-141:

“We use the Darwin code⁵⁴, which neglects the transverse displacement current but leaves the physics of whistler mode waves unaffected. It removes the time-step constraint of the full electromagnetic code, and the computational efficiency can be greatly improved by using a large time-step^{55,56}.”

References:

An, X., et al. (2017a) Electrostatic and whistler instabilities excited by an electron beam, Phys. Plasmas, <http://dx.doi.org/10.1063/1.4986511>

An, X., et al. (2017b), On the parameter dependence of the whistler anisotropy instability, J. Geophys. Res. Space Physics, 122, 2001–2009, doi:10.1002/2017JA023895.

g) It would be useful to show the dynamic spectra of chorus emissions from simulations during various stages in simulation and compare simulation results with those observed.

Reply: The spectrogram in Figure 4 shows the chorus spectra in (ω, k) space during three stages. The observation of one-band chorus that straddles $0.5 f_{ce}$ is rare. In the new Figure 5b, we showed a pure one-band chorus associated with one anisotropic electron component, and a two-band chorus waves associated with two anisotropic electron components.

h) Missing references: for example, references for bi-Maxwellian distribution, Kappa distribution, Landau resonant energy calculations, Box-Muller method are missed. References for the validity of the models used in the paper are missing.

Reply: Thank you for this note. We have cited several papers describing the bi-Maxwellian distribution and Kappa distributions and the Box-Muller method, and added a section describing the calculation of representative Landau resonant velocity as seen on Lines 253-262:

“Estimation of Landau resonant velocity. The Landau resonant velocities in this study is calculated based on cold plasma theory⁴⁷. The refractive index n of a parallel propagating chorus wave is

$$n^2 \simeq 1 + \frac{\omega_{pe}^2}{\omega(\omega_{ce} - \omega)}.$$

For $\omega = 0.5 \omega_{ce}$, the Landau resonant velocity is

$$v_{Landau} = \frac{\omega}{k_{\parallel}} = \frac{c}{n} \simeq c / \sqrt{1 + 4\omega_{pe}^2/\omega_{ce}^2}.$$

In the cases of this study, the ω_{pe}/ω_{ce} ratio is generally larger than 3, therefore $v_{Landau} \simeq c\omega_{ce}/2\omega_{pe} = 0.5 v_{Ae}$ where v_{Ae} is the electron Alfvénic velocity. The Landau resonant velocities for quasi-parallel propagating banded chorus waves are typically within a narrow range around this representative velocity (Figure 2b).”

Minor comments (not extensive):

1. Lines 47-52: It will be useful to give Van Allen Probe orbit information, L-shell, MLT at which chorus was observed, measured f_{uh} , calculated values of f_{pe} and f_{ce} .

Reply: We have added the Van Allen Probe orbit information in Figure 2b now. We also added the number of f_{pe} and f_{ce} in the manuscript on Lines 103.

2. Line 53, Are the f_{pe} and f_{ce} values along the orbit roughly constant?

Reply: We deleted this sentence now.

3. The values of f_{pe}/f_{ce} used for calculating resonant energies (Figure 2; $f_{pe}/f_{ce}=5.5$) and particle in cell simulations ($f_{pe}/f_{ce}=5$) was different why?

Reply: We use a slightly smaller f_{pe}/f_{ce} ratio to increase the number of k mode of chorus waves. This does not change the basic excitation mechanism of banded chorus waves.

4. The discussion section is more like summary and concluding remarks

Reply: To make this section containing more discussions, we moved the two paragraphs above, which mainly discuss the differences between the simulation and the real case, to the Discussion section.

5. Organization of the paper: It will help the reader if the methods for calculating growth rates and fitting electron distributed is presented earlier in the simulations section rather than at the end of the paper.

Reply: In the original manuscript, those methods are described in the “Methods” section at the end of the paper, so as to meet the NC specifications. However, we agree that it will help the reader if we describe the methods before the simulations results. In the revised manuscript, we briefly described the fitting procedure in the main text as space would allow, while the details are presented in the “Method” section. The description of the linear growth calculation method is moved to the main text.

6. How do the authors explain the variability of the banded chorus gap width?

Reply: Our explanation is that the gap width is possibly associated with the electron distributions. At the beginning of Landau resonant acceleration and the separation of the two anisotropic electrons, the spectral gap is presumably small. As the Landau resonant acceleration continues and the separation of two anisotropic components deepens, a wider spectral gap may develop. Figure R5 shows the wave spectra and electron distribution for the Event #1 and #2 in our statistical study. The 2013-01-11 event shows a wide spectral gap in association with two separated anisotropic components, while the 2013-01-12 event shows a narrow spectral gap,

possibly because the Landau resonant acceleration and the separation of two electron components are in the initial stage. While we have a basic understanding of how this works, we plan to investigate it more deeply and address this issue in a future paper.

Figure R5. The chorus wave spectra and the concurrent electron PSD for Event #1 and #2 in the statistical study.

References:

Burtis, W. J. and Helliwell, R. A. (1976) Magnetospheric chorus: occurrence patterns and normalized frequency. *Planet. Space Sci.* 24, 1007.

Hartley, D. P., Kletzing, C. A., Chen, L., Horne, R. B., & Santolík, O. (2019). Van Allen Probes observations of chorus wave vector orientations: Implications for the chorus-to-hiss mechanism. *Geophysical Research Letters*, 46, 2337–2346. <https://doi.org/10.1029/2019GL082111>

Li, W., O. Santolik, J. Bortnik, R. M. Thorne, C. A. Kletzing, W. S. Kurth, and G. B. Hospodarsky (2016), New chorus wave properties near the equator from Van Allen Probes wave observations, *Geophys. Res. Lett.*, 43, 4725–4735, doi:10.1002/2016GL068780

Schrifer, D., et al. (2010), Generation of whistler mode emissions in the inner magnetosphere: An event study, *J. Geophys. Res.*, 115, A00F17, doi:10.1029/2009JA014932.

Reply: We have cited all these papers in the revised manuscripts.

Reply to comments by Reviewer #3

Reviewer #3 (Remarks to the Author):

In this manuscript, the authors at first observed that two-band structure of chorus waves are usually associated with two anisotropic electron components, and then they tried to use 1-D PIC simulations to demonstrate that the initially excited single-band chorus waves can alter electron distribution via Landau resonance and divides the electron anisotropy into a low and high energy components, and at last both the upper- and low-band chorus waves are excited. The idea is attractive, however, I think that the evidence is not enough to support the conclusion. The reasons are described as follows.

1. After I read the manuscript, I am a little confused on where the two-band structure of chorus waves is formed. From the Figure 1, the waves have obvious rising tone, so they have already left the source region of the excited waves. However, in the simulation, the authors are describing the excitation of whistler waves by an electron temperature anisotropy, it is in the source region of the excited waves.

Reply: Thank you, this is an excellent question. The chorus waves are indeed generated in the source region, that is, at the magnetic equator where the electron anisotropy peaks. The observed chorus waves shown in Figure 1 were at a Latitude of 6.2° which is off the equator but not far away from the source region. We therefore slightly increased the electron anisotropy in the PSD fitting process, so that the fitted distribution better represents the distribution at the source region. In the PIC simulation, we also used a higher anisotropy ($T_{\perp}/T_{\parallel} = 4$) to model the initial electron distribution before the excitation of chorus waves.

2. In the observations, the chorus waves have a two-band structure, and the amplitude of the lower band is larger than that of the upper band. However, from Figure 2, the growth rate of the upper band is larger than that of the lower band. If the two bands of the chorus waves are excited separately by a low and high energy electron components, it is difficult to explain it. Again, if according to the electron distribution the whistler waves have a high growth rate, they should be in the source region. An electron temperature anisotropy should excite whistler waves propagating both parallel and antiparallel to the magnetic field, then how can the waves have an obvious rising tone.

Reply: The author is quite correct in their assessment but we note that this paper addresses the generation mechanism of two-band chorus structures and is less concerned with the power distribution between the bands (though this is important for scattering and other reasons of course). The PIC simulation addresses the key process in forming two-band chorus waves, i.e., the separation of anisotropic electrons into two components caused by Landau resonant acceleration. However, there are several differences between the observed electron distribution and the modeled one. The observed electron distribution consists of 5 components electron components, and the electron distribution had already been altered by the chorus waves. In the

PIC simulation, however, we only use two components to roughly model the electron distribution before the excitation of chorus waves. Moreover, in reality, the electron density is mainly contributed by the cold component ($\sim 92\%$ in this case, $v_{te} \approx 1000$ km/s). However, in the model, the cold component ($v_{te} \approx 3000$ km/s) contributes 80% of the total density, while the warm component contributed 20%. We successfully simulated the separation of two anisotropic electron components and the generation of two-band chorus waves. The intensity of the two bands can change as the wave-particle interaction continue. Which band has larger intensity is not the key issue.

The whistler mode wave spectra in Figure 4 are shown as a function of frequency and k mode number. We illustrated in Figure 4b that the k mode number represents the number of wavelengths in the entire simulation space. We added a sentence on Lines 160-162:

“Note that the wave spectra are shown as a function of frequency (ω) and k mode number (the number of wavelengths in the simulation space), and the rising-tone structure is not simulated here.”

The mechanism of the rising-tone structure is one major problem in nonlinear wave-particle interactions, and is beyond the scope of this study.

3. In simulations of whistler waves excited by an electron temperature anisotropy, during the relaxation of the temperature anisotropy, the frequency will drift to a value. It is not strange to observe a gap due to Landau resonance if the excited waves obliquely propagate. So, it is not clear whether the low-band of whistler waves are excited by the high energy component.

Reply: The whistler mode waves in the Earth’s inner magnetosphere are generated from anisotropic electrons through cyclotron resonance (e.g., Kennel & Petschek, 1966; Schriver et al., 2010; Li et al., 2010). The minimum resonant energy versus wave frequency under the condition of $f_{pe}/f_{ce}=5.5$ is shown in Figure R6. During the process of cyclotron resonance, the lower-band chorus waves interact with high-energy electrons, while the upper-band chorus waves interact with low-energy electrons.

Figure R6. Minimum resonant energies for cyclotron and Landau resonances as a function of wave frequency for various wave normal angles under the plasma condition of $f_{pe}/f_{ce} = 5.5$.

Reviewers' Comments:

Reviewer #1:

Remarks to the Author:

The reviewer thinks that the authors have considerably improved the manuscript, and the reviewer now believes that the manuscript is acceptable for publication.

Reviewer #2:

Remarks to the Author:

Reviewer appreciates that authors have carefully addressed the points raised in the review.

Reviewer #3:

Remarks to the Author:

I don't think that the answers of the authors have convinced me. I reiterate my two serious points.

1. Obviously, the simulations concern the excitation of whistler waves, which is in the source region of whistler waves. However, I wonder whether the observations are in the source region of whistler waves. If the observations are not in the source region of whistler waves. Then, the authors cannot compare the simulations with the observation. If the authors want to convince the readers that their observations are in the source region of whistler waves, they should give the directions of Poynting fluxes of the whistler waves. If the whistler waves are in their source region, the Poynting vectors should have components both parallel and anti-parallel to the magnetic field.
2. The authors also cannot convince me that the low-band and high-band whistler waves are excited by two separate electrons components with temperature anisotropy. During the relaxation of the temperature anisotropy, the frequencies will also drift to lower value. I think that the authors should run another case, and the whistler waves propagate along the magnetic field. If the spectra of the whistler waves are similar to that of oblique whistler waves, except without a gap around 0.5 , then the low-band whistler waves should be the results from a quasi-linear process during the relaxation of the temperature anisotropy, not excited by another component with a temperature anisotropy.

Response to Reviewers' comments

Reviewer #1 (Remarks to the Author):

The reviewer thinks that the authors have considerably improved the manuscript, and the reviewer now believes that the manuscript is acceptable for publication.

Reviewer #2 (Remarks to the Author):

Reviewer appreciates that authors have carefully addressed the points raised in the review.

Reviewer #3 (Remarks to the Author):

I don't think that the answers of the authors have convinced me. I reiterate my two serious points.

1. Obviously, the simulations concern the excitation of whistler waves, which is in the source region of whistler waves. However, I wonder whether the observations are in the source region of whistler waves. If the observations are not in the source region of whistler waves. Then, the authors cannot compare the simulations with the observation. If the authors want to convince the readers that their observations are in the source region of whistler waves, they should give the directions of Poynting fluxes of the whistler waves. If the whistler waves are in their source region, the Poynting vectors should have components both parallel and anti-parallel to the magnetic field.

Reply: Recent studies have shown banded chorus wave observations at equator, indicating that the spectral gap is formed at the source region (See Figure 1 in Fu et al., 2014, doi:10.1002/2014JA020364). Below we show another case where the banded chorus waves are exactly observed at the source region, as can be identified by the coexistence of parallel and anti-parallel propagating emissions. The electron distribution shows two anisotropic components (80-200 eV and >500 eV), while the electrons at ~300 eV, which is the estimated Landau resonant energy, show a clear evidence of parallel acceleration. The event studied in this manuscript were measured at a latitude of -6° , however, it does not change our understanding of the excitation of banded chorus waves by mapping the measured electron distribution to the equatorial distribution. We therefore do not change our event study.

Figure R1. An example of two-band chorus waves observed at the source region where the parallel propagating and anti-parallel propagating emissions coexist. The electron distribution shows two anisotropic components separated by a parallel acceleration at ~300 eV.

2. The authors also cannot convince me that the low-band and high-band whistler waves are excited by two separate electrons components with temperature anisotropy. During the relaxation of the temperature anisotropy, the frequencies will also drift to lower value. I think that the authors should run another case, and the whistler waves propagate along the magnetic field. If the spectra of the whistler waves are similar to that of oblique whistler waves, except without a gap around 0.5, then the low-band whistler waves should be the results from a quasi-linear process during the relaxation of the temperature anisotropy, not excited by another component with a temperature anisotropy.

Reply: The statistical study in this manuscript have clearly shown that the banded chorus waves are commonly accompanied with two anisotropic electron components separately by the Landau resonant electrons in between. The resonance energy analysis (Figure 2b in the manuscript) and the linear growth calculation (Figure 2c) prove that the lower band and upper band whistlers are excited by a high energy and a low energy anisotropic electron components, respectively.

Below we show a simulation result by setting the normal angle to be 0° while the rest of parameters unchanged. We see that the excited waves are similar to that shown in the manuscript, except that there is no spectral gap at 0.5 fce. This is because parallel propagating whistler mode waves cannot accelerate electrons via Landau resonance. In the PIC simulation, only the integer k modes ($k=1,2,3,\dots$, the corresponding wavelengths are $L, L/2, L/3\dots$ where L is the overall spatial length in the simulation) are simulated, but non-integer modes and $k<1$ modes cannot be simulated. As a result, the anisotropic electrons at the high-energy end are not relaxed due to the suppression of wave growth at small k modes (e.g., $k=0.5, k=1.5$). In reality, the electrons at the tail should also be significantly relaxed to excite whistlers with small k mode numbers (which correspond to long wavelengths and low frequencies)

Figure R2. The PIC simulation results of parallel whistler mode wave growth from anisotropic electrons. The spectral gap at $0.5 f_{ce}$ is not formed in this case because the parallel propagating whistler mode waves cannot accelerate electron via Landau resonance.

Reviewers' Comments:

Reviewer #3:

Remarks to the Author:

The authors have answered my concerns carefully, and I think that it is ready to be accepted.